# Structural Effects of Mass Distributions in a Floating Photovoltaic Power Plant

Chun Bao Li [1,2,3] and Joonmo Choung [4,*]

1    Key Laboratory of High Performance Ship Technology (Wuhan University of Technology),
     Ministry of Education, Wuhan 430063, China
2    School of Naval Architecture, Ocean and Energy Power Engineering, Wuhan University of Technology,
     Wuhan 430063, China
3    Weihai Research Institute of Wuhan University of Technology, Weihai 264200, China
4    Department of Naval Architecture and Ocean Engineering, Inha University, Incheon 22212, Korea
*    Correspondence: heroeswise2@gmail.com; Tel.: +82-32-860-7346

**Abstract:** This study deals with a solar photovoltaic demonstration project composed of four types of sub-plants that will be operated in the Saemangeum Seawall coast. The project aimed to investigate the most efficient sub-plant types. Hydrodynamic analyses were undertaken to obtain the loads exerted on the floating photovoltaic power plants on which two kinds of frame structures supported shed- and gable-type photovoltaic panels, producing the four types of sub-plants composed of three floaters. Hydrodynamic interactions between the floaters were considered because floaters were linked with hinge joints. The pressure and acceleration response operator amplitudes were transferred to the finite element analysis model using an in-house code. Because each sub-plant had a different mass and second moments of mass, it was found that huge stresses had been retained in hinge joints. After the masses in the twelve floaters were evenly distributed, the maximum stresses were reduced so that they were less than material yield strengths. There were larger stresses in the POSCO (Pohang Iron and Steel Company) magnesium alloy coating (POSMAC) frames than in the fiber-reinforced plastic (FRP) frames because the POSMAC frame had an open-channel section. It is concluded that weight in each floating unit should be evenly controlled if hinged joints are used to link the floaters.

**Keywords:** floating photovoltaic power plant; sub-plant; floater; frame structure; response operator amplitude

## 1. Introduction

Recently, national carbon-free policies have been realized around the world. Social attention to eco-friendly renewable energy is continuously increasing, and the solar power generation market is gradually developing. According to the World Bank report [1], if 1% of the world's reservoir water surface is used, the capacity of floating photovoltaic (PV) power generation is projected to be 400 GW. Its efficiency is estimated to be 5–15% higher than that of land solar power. The International Energy Agency (IEA) survey [2] showed that the global solar power installed capacity in 2019 was 114.9 GW, including 30.1 GW in China, 16.0 GW in the EU, and 13.3 GW in the US. The BP statistical data [3] showed that global solar power generation in 2019 recorded 724.1 TWh, an increase of 24.3% compared to the same period of the previous year.

While various problems have been raised about land PV power plants, such as that solar power generation requires a large-scale installation space to meet the planned power generation, floating PV power using public waters can be a good alternative to the land space problems. As shown in Figure 1, the power generation method using floating PV structures is similar to the land-based one, but there are some differences in terms of foundation structures and associated station-keeping methods.

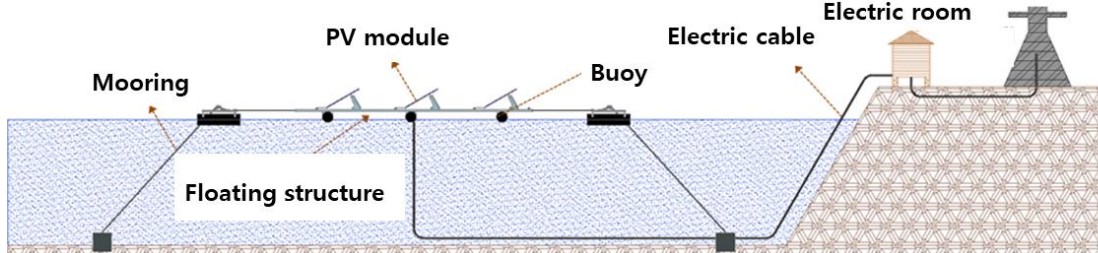

**Figure 1.** A schematic of floating PV power generation system.

The floating PV structural system consists of frame structures supporting PV modules, buoys, and mooring systems (see Figure 1). A mooring system includes mooring lines, lines stoppers at floater attachment points (usually called fairlead point even though no fairleads are used), and anchors to fix horizontal and vertical displacements. In order to keep the initial position of the power plant under environmental loads, it is necessary to maintain effective tensile forces on the mooring lines at all times. Therefore, it is required to develop a technology to design the configuration of mooring lines.

While some live and dead loads dominantly apply to land-based PV power generation systems, floating PV power generation systems are largely affected by land-based loads and hydrodynamic loads. It is necessary to develop an advanced structural system that can maintain durability and safety in marine environments such as wetting and salt corrosion. Floating PV structures require an advanced technological background including hydrodynamics and structural dynamics under an in situ environment. The associated international and national rules and codes have been released [4–6]. Recently, the Det Norske Veritas (DNV) developed a design guideline (DNV-RP-0584) dealing with design considerations for the floating PV structures [7]. As a result of a joint development project, this guideline covers all topics associated with floating PV platforms including marine environmental conditions, platform design and analysis technologies, buoy and mooring facilities, installation, operation, maintenance, decommissioning, safety, and the estimation of levelized ground cost of electricity (LCOE).

It is necessary to obtain synthetic rope mooring material properties for floating PV structures. Mechanical properties of a synthetic fiber chai were provided by Kim et al. [8] and Chung et al. [9]. Kim et al. [10] performed fluid–structure interaction (FSI) analysis using finite element method and finite volume method to examine the behavior of a floating PV platform composed of PV support frames and buoys made from fiber-reinforced plastic (FRP) and high-density polyethylene (HDPE), respectively. Considering wave height and wave period in various methods based on linear wave theory, it was found that maximum stress occurred in vertical and horizontal members at the front edge when the wave crest passes through the structure. Kim et al. [11] performed wave load analysis on floating bodies composed of concrete block structures to verify the strength of the connection part of a modular floating structure that can be applied to a floating PV power generation. Using hydrodynamic forces, stresses were evaluated and compared by dynamic analysis of connection parts based on beam theory. Li et al. estimated motion performance, dominant load parameters, and hydrodynamic forces of a floating PV structure based on design wave method [12]. Using hydrodynamic forces, he evaluated yield strength from the entire structural analysis. Yoon et al. [13] carried out design verification based on structural analysis for a floating PV structure made of FRP composite material suggested by Lee et al. [14]. Lin et al. [15] examined the correlation between energy efficiency and tilt angle to apply the decommissioned floating production storage and offloading (FPSO) unit to a floating PV structure. Performing frequency-domain motion analysis of the FPSO, it was confirmed that roll motion had a much more negative effect on total radiation than pitch motion. Studies regarding mooring design have been released, even though they were for the different types of floating renewable platforms such as floating wave converters and wind turbines [16,17].

A pilot PV power generation system, which is called hereafter floating PV power plant, is discussed in this paper in terms of structural integrity based on hydrodynamic and structural dynamic simulations. It will be operated in Saemangeum Seawall Lake, South Korea. It consists of four sub-plants composed of three floater units. Because four sub-plants have different mass distributions, the purpose of this paper is to analyze the effect of mass distributions on structural integrity.

## 2. Geometric Details of the Pilot PV Platform

As shown in Figure 2a, the pilot floating PV power plant is composed of four sub-plants, whereas a sub-plant includes three floater units. Twelve floater units were linked with hinges. The PV modules were arranged in the shed (SHD) and gable (GBL) types (refer to Figure 2a,b) and supported by two different profiles made from fiber-reinforced plastic (FRP) and formed sheet steel called POSMAC (POS) (see Figure 2c). The FRP frame has an I-beam section as delineated in Figure 2c, while the section of the POS frame is channel type.

Six buoys per floater made from polyethylene (PE) support the frame structures. Seventy-two buoys were used in total. The configuration of a buoy is shown in Figure 2a,b. Four synthetic mooring lines were arranged at each side of the PV power plant where two mooring lines were anchored by a heavy concrete block (refer to Figure 2e). Table 1 lists the detailed specifications of the mooring lines and the Figure 2f presents the layout of the mooring lines.

**Table 1.** Properties of mooring systems.

| Items | Unit | Value |
|---|---|---|
| Nominal diameter | m | 0.024 |
| Minimum breaking load | kN | 57.0 |
| Mooring density | kg/m | 0.27 |
| Unstretched length | m | 9.513 |
| Young's modulus | MPa | 1800 |
| Axial stiffness | N/m | $6.53 \times 10^4$ |

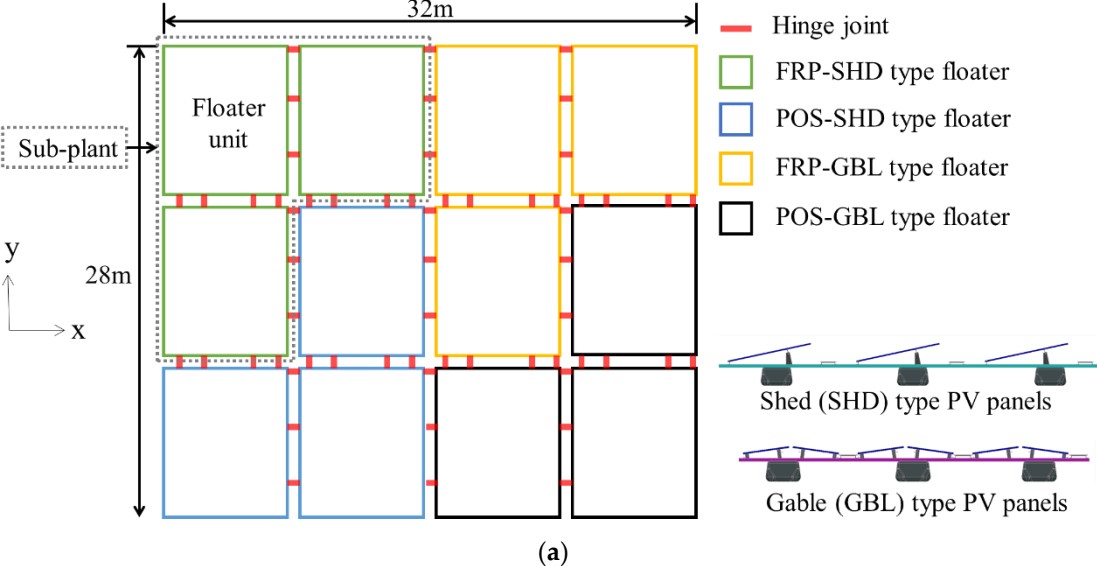

(**a**)

**Figure 2.** *Cont.*

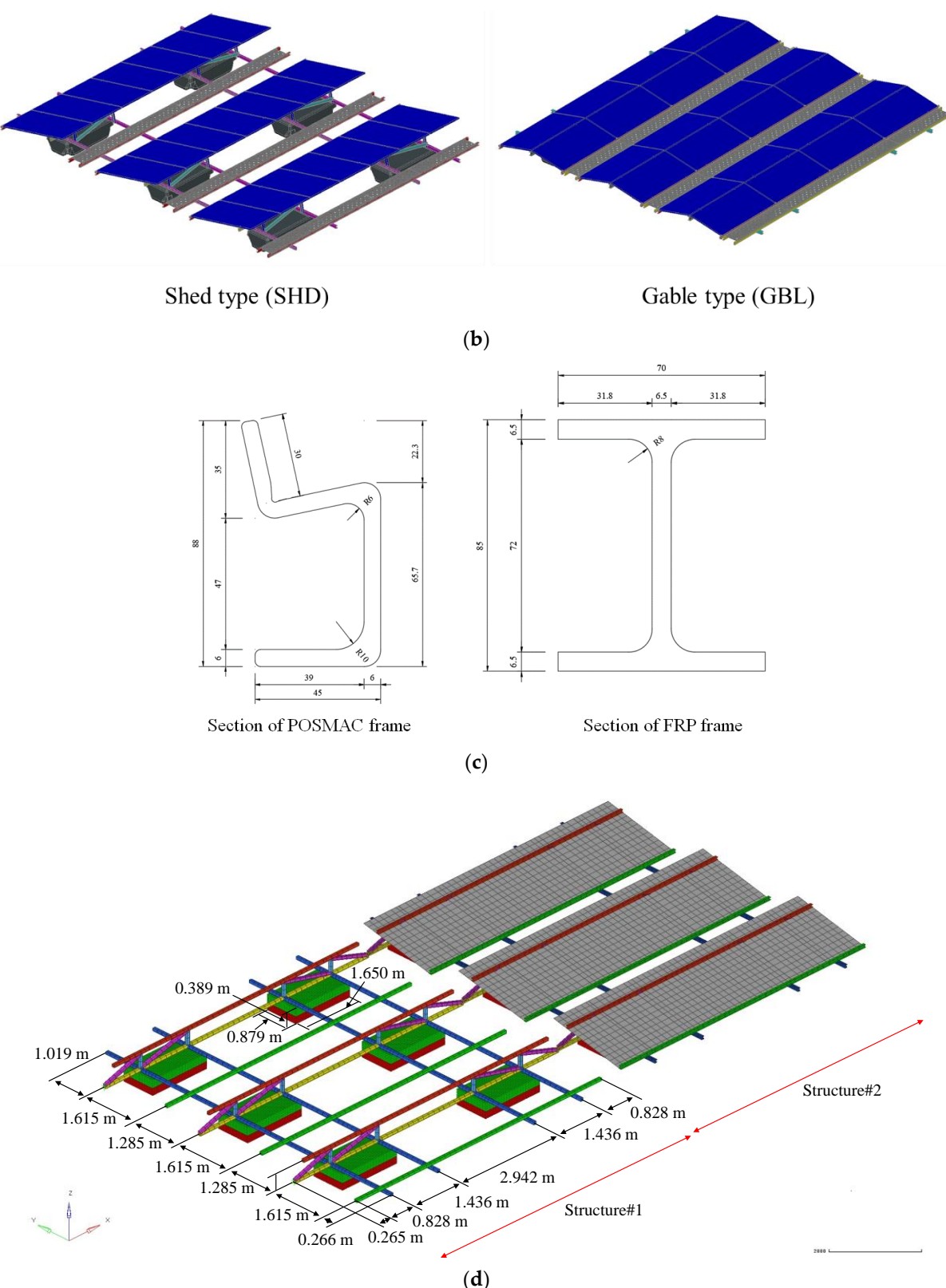

Shed type (SHD)　　　　　　　　　　　　　　　　Gable type (GBL)

(**b**)

Section of POSMAC frame　　　　　　　　　　　Section of FRP frame

(**c**)

(**d**)

**Figure 2.** *Cont.*

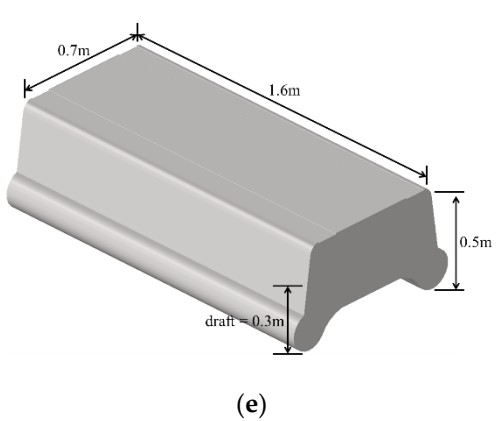

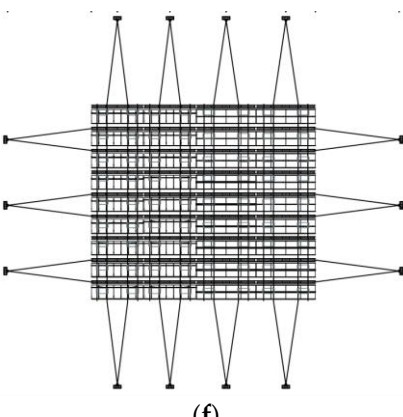

(e)                (f)

**Figure 2.** Configuration of the floating PV platform composed of 12 floating units: (**a**) Sketch of a plan view of overall PV power plant; (**b**) PV arrangement; (**c**) Sections of FRP and POS frames; (**d**) Frame structure of a shed-type floater unit; (**e**) Buoy; (**f**) Mooring arrangement.

## 3. Hydrodynamic Analyses

### 3.1. Theoretical Background

When a floating body moves due to incident waves, the added mass-induced force should be estimated. The moving body also can radiate away waves dissipating some parts of kinetic energy. This energy dissipation-induced force is usually called wave damping force. One of the most important objectives in the hydrodynamic frequency response analysis is to obtain the added mass $m$ and wave damping coefficient $b$ using radiation potential $\phi$ (refer to Equation (1)).

Meanwhile, wave excitation forces due to incident wave and diffracted wave should be obtained from the hydrodynamic frequency response analysis. The incident wave potential $\phi_b$ and diffraction wave potential $\phi_d$ are used to obtain the wave excitation force $F_w$ using Equation (2). In this study, Airy's linear wave theory was introduced, as shown in Equation (3), where wave elevation $\eta$ is given as a cosine function.

$$m - \frac{i}{\omega}b = \rho \iint \boldsymbol{n}\cdot\boldsymbol{\phi}\,dS \tag{1}$$

$$\boldsymbol{F_w} = i\omega\rho \iint \boldsymbol{n}\cdot(\boldsymbol{\phi_i} + \boldsymbol{\phi_d})\,dS \tag{2}$$

$$\eta(x,t) = \frac{H}{2}\,cos(kx - \omega t) \tag{3}$$

$i$ : complex number;
$\omega$ : wave frequency;
$S$ : wetted surface;
$\boldsymbol{n}$ : normal vector to wetted surface;
$x$ : coordinate in incident wave direction;
$t$ : time.

### 3.2. Hydrodynamic Analysis Model

A commercial potential flow code Ansys/Aqwa [18] was used to perform hydrodynamic analyses under the incident wave loadings from three directions: 0° (x-direction), 45°, and 90° (y-direction). Figure 3 shows the panel elements for the buoys and PV modules. Wetted structures are necessary for the hydrodynamic analysis under wave loadings, but the PV modules were included for the structure analysis application of wind-induced drag forces. The support frames were not included in the hydrodynamic model because buoys and PV modules were considered to be a single rigid body. The number of diffraction and non-diffraction panel elements are summarized in Table 2.

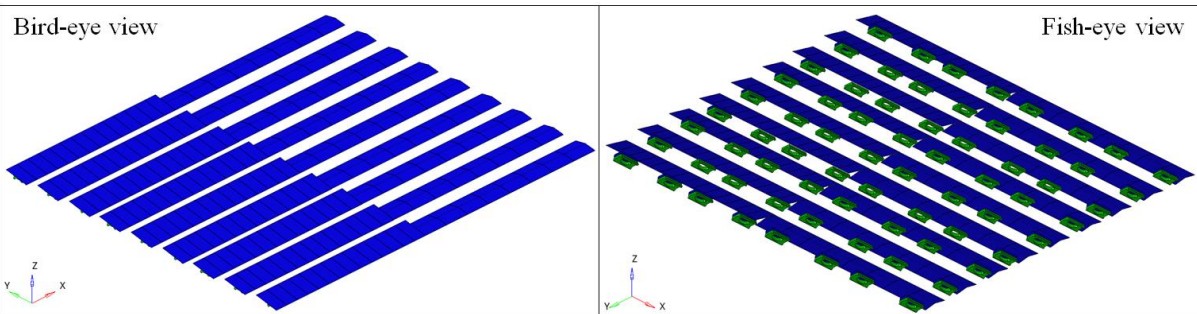

**Figure 3.** Configuration of the floating PV plant composed of four sub-plants.

**Table 2.** Summary of panel elements.

| Member | Number of Wet Panels (Diffraction) | Number of Dry Panels (Non-Diffraction) |
|---|---|---|
| PV module | 0 | 270 |
| PE buoy | 21,312 | 6360 |
| Total | 21,312 | 6630 |

It is very reasonable that the mass distribution was uneven in the four sub-plants because each was supported by different frames and has different PV module arrangements. This uneven mass distribution would elevate shear loads between the sub-plants which can induce bending moments at the hinge joints. Therefore, as listed in Table 3, two cases were considered: M1 with uneven mass distribution and M2 with even mass distribution. It was very difficult to reduce the weight of a heavier sub-plant, but it was easy to increase the weight of a lighter sub-plant. Therefore, the mass of all sub-plants was modified to be the same as that of the heaviest sub-plant. $I_x$, $I_y$, and $I_z$ are the second moments of mass with respect to three directions.

**Table 3.** Mass information according to two mass distribution cases.

| Member | | M1 (Uneven Mass Dist.) | M2 (Even Mass Dist.) |
|---|---|---|---|
| | Sub-plant FRP-SHD | 2817.4 | 3934.4 |
| Displacement (kg) | Sub-plant POS-SHD | 3254.6 | 3934.4 |
| | Sub-plant FRP-GBL | 3241.0 | 3934.4 |
| | Sub-plant POS-GBL | 3934.4 | 3934.4 |
| Center of mass in z-direction (m) | | 0.4083 | 0.3750 |
| $I_x$ (kg-m$^2$) | | $8.6481 \times 10^5$ | $1.0284 \times 10^6$ |
| $I_y$ (kg-m$^2$) | | $1.1313 \times 10^6$ | $1.3394 \times 10^6$ |
| $I_z$ (kg-m$^2$) | | $1.9910 \times 10^6$ | $2.3669 \times 10^6$ |

In this study, it was not necessary to use many frequencies because a single design wave was used, but a wide range of wave frequencies (0.1–7 rad/s) was used to observe motion response amplitude operators (RAOs) comprehensively (refer to Table 3). The number of frequencies was forty and ranged from 0.1 rad/s to 7.0 rad/s. Pastor and Liu [19] presented that the number of frequencies was appropriate to adequately represent most wave spectrums, and the range of frequencies was wide enough to avoid energy loss in most wave spectrums. In order to consider the viscous damping effect for roll response especially, some experimental or computational fluid dynamics simulations should be performed to obtain the relative viscous damping ratios. Consideration of viscous damping effect should reduce the motion, velocity, and acceleration of the real floating photovoltaic power plant [20]. It leads to more conservative structural analysis results. However, the

viscous damping was not taken into account in this study. Environmental conditions such as water depth and seawater density are summarized in Table 4.

**Table 4.** Conditions for hydrodynamic frequency response analysis.

| Member | Value |
|---|---|
| Water depth (m) | 4.0 |
| Seawater density (kg/m$^3$) | 1025.0 |
| Gravity (m/s$^2$) | 9.81 |
| Wave frequency range (rad/s) | 0.1–7.0 |
| Number of wave frequency | 40 |
| Viscous damping ratio | 0.0 |

### 3.3. Hydrodynamic Analysis Results

Figure 4 represents the wave elevation distribution against each incident wave angle where the diffracted waves are observed out of the global PV power plant as well as between the floater units. The trapped-wave modes between the inside floating units are highly developed when the incident waves are following and beam seas (see Figure 4a,c). The reason is that the front buoys are aligned with the rear buoys. In the case of 45°, the trapped waves hardly occur because the front and rear buoys are not perfectly aligned.

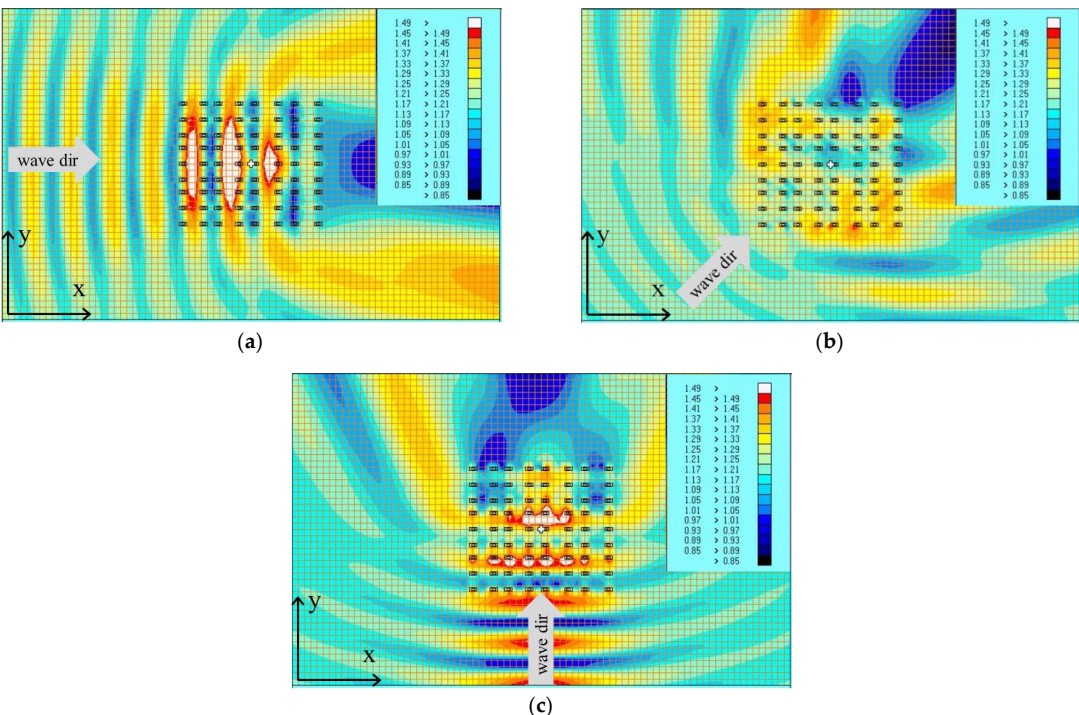

**Figure 4.** Wave elevation contour: (**a**) incident angle = 0°; (**b**) incident angle = 45°; (**c**) incident angle = 90°.

If the RAOs for the twelve floaters are collected into a graph, they become too complicated. For this reason, assuming the twelve floaters as a single body, additional hydrodynamic frequency response analysis was performed, and the results are presented in Figure 5 for the more readable motion RAO presentation. The heave, roll, and pitch motion RAOs are presented in Figure 5. The roll motion was not excited by the following sea (angle = 0°), while the beam sea (angle = 90°) can slightly develop the pitch motion at a quite high frequency of 4 rad/s. As expected, the quartering sea (angle = 45°) excited heave, roll, and pitch motions, simultaneously.

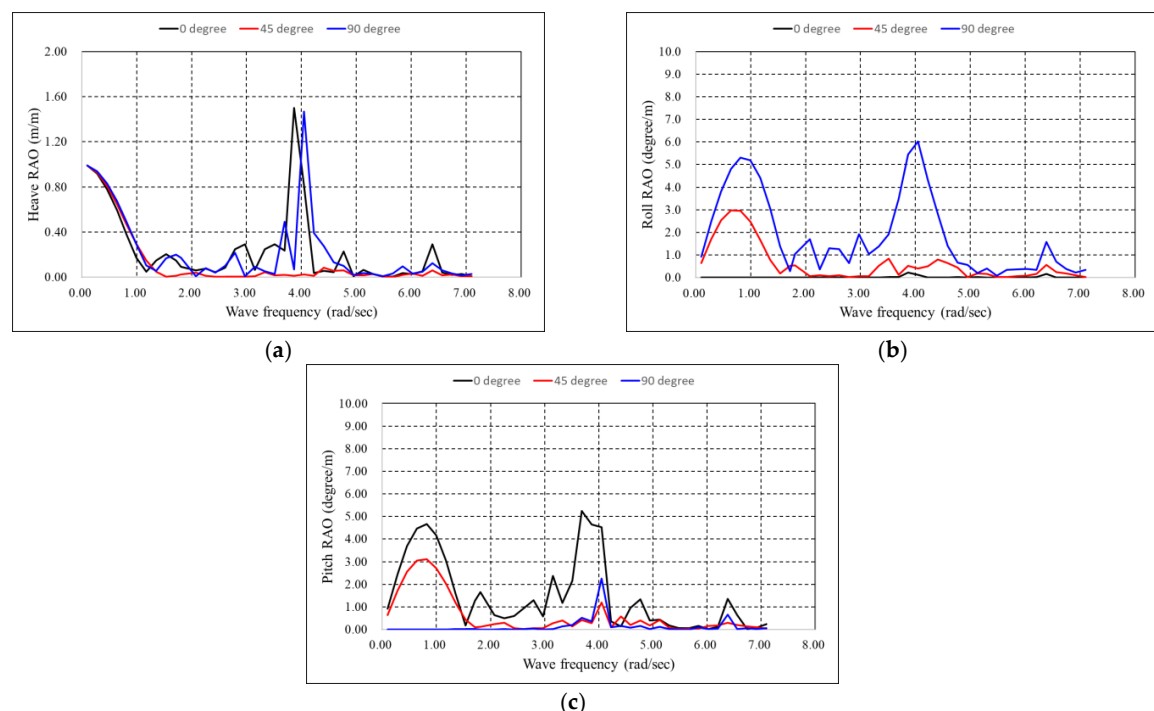

**Figure 5.** Motion RAOs: (**a**) heave; (**b**) roll; (**c**) pitch.

The impulse response functions for heave, roll, and pitch motion were listed in Figure 6, which represents the wave radiation damping force at the time *t* resulting from a unit impulse in velocity at the time zero. Then, the wave radiation damping force decays to zero at 30 s. The convolution between the impulse response functions and velocity gives wave loads to structural analysis.

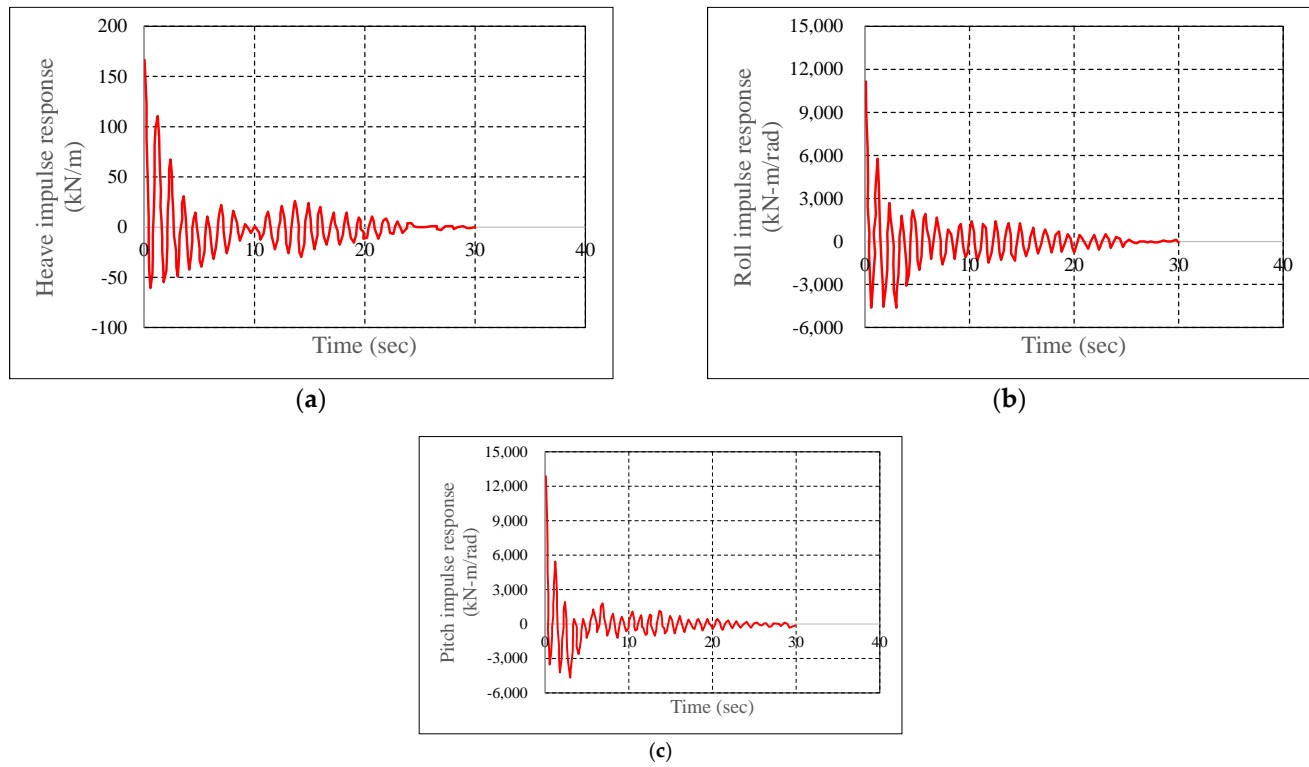

**Figure 6.** Impulse response functions: (**a**) heave; (**b**) roll; (**c**) pitch.

## 4. Structural Analysis

### 4.1. Structural Analysis Model

The PE buoys were modeled using solid elements, while the beam elements were used for the frame structures. The solar modules were presented by elastic shell elements. Structural analysis models were generated on the basis that twelve floater units were connected with the pin-type hinge joint elements. The hinge joint elements play a very important role to allow relative rotations between successive floater units so that generation of the bending moments between the floater units could be minimized. The kinematic coupling elements to bond the buoys and frames avoid probable stress concentration at the hinge joints [21,22]. In reality, the joints were quite firmly fastened with a couple of bolts and nuts. We assumed that the fairlead points at which mooring lines are tied were supported by grounded spring elements [23–26]. One end of the ground spring element is fixed to the earth and the other end is connected to the structure. The spring elements had three degrees of freedom in three directions. A very small stiffness was uniformly assigned to the spring elements to avoid rigid body motion in the structural analyses. Therefore, no boundary conditions were applied, and the inertia relief technique was not employed. The thickness of the PV modules was assumed to be 5 mm from the reference survey [27]. The element information is summarized in Table 5 with the number of elements used. The B31 is a 3D Timoshenko beam element with 6 degrees of freedom. The S4 is a shell element with a full-integration scheme with five degrees of freedom. The last degree of freedom provides drilling stiffness. The C3D8 is a hexahedron solid element with a full-integration scheme with three degrees of freedom. They can be used with elastic or elastic–plastic material properties, but only elastic material properties were assigned to the elements in this study.

**Table 5.** Element information for structural frequency response analysis.

| Member | Types of Elements | Number of Elements |
|---|---|---|
| Frame | Beam (B31) | 11,955 |
| PE buoy | Solid (C3D8) | 30,6175 |
| PV module | Shell (S4) | 3456 and 59,166 |
| Hinge joint | Pin-type MPC | 86 |
| Fairlead point | Spring | 140 |
| Bonding bet. frame and buoy | Kinematic coupling | 1476 |
| Sum | | 382,454 |

The structural analysis model was built as shown in Figure 7, where some PV modules were intentionally masked to provide better visibility of the FEA model. The density, elastic modulus, Poisson ratio, and yield strength of each material are delineated in Table 6. Some of them were taken from web searches [28]. The yield strength of the POSMAC ranged broadly from 200 MPa to 600 MPa; thus, the highest one was assumed in this study. The density of the M2 model which has even mass distribution was scaled up uniformly for all materials until the masses specified in Table 3 were obtained.

The panel pressure RAOs were transferred to the finite elements using Equation (4) and (5) where $p_i$ and $p_j^w$ are $i^{th}$ panel pressure and weighted pressure exerted to $j^{th}$ finite element. $\boldsymbol{n}$ denotes the number of panels sharing edges with the panel that overlays the most with the considered finite element. The weighting exponent $\alpha$ is used to increase weighting factor $w_i$ at $i^{th}$ panel. The weighting factor is inversely proportional to the ratio of distances. If the distance from $i^{th}$ panel to $j^{th}$ finite element is close, the weighting factor approaches unity based on Equation (5). In this study, the pressure transfer was performed using an in-house code developed by the authors [29].

Still water buoyancy and gravity loads were applied to the wet elements and all elements, respectively. Acceleration RAO components were also transferred to the FEA model where the rotation components were applied with respect to the center of mass (COM).

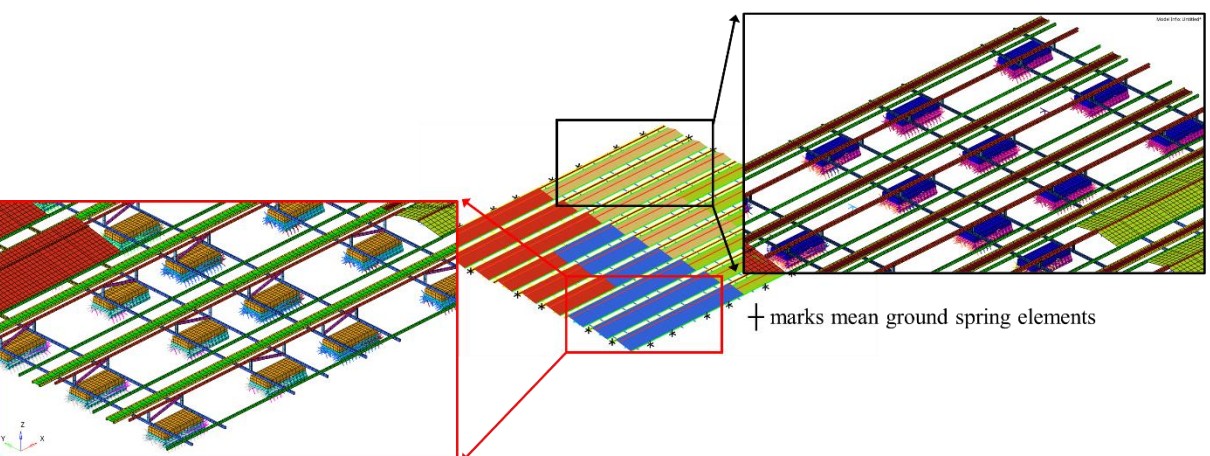

**Figure 7.** Structural analysis model.

**Table 6.** Material information for structure analysis.

| Member | Density (M1) (kg/m$^3$) | Elastic Modulus (GPa) | Poisson Ratio | Yield Strength (MPa) |
|---|---|---|---|---|
| FRP Frame | $1.878 \times 10^2$ | 25 | 0.4 | 600 |
| POSMAC frame | $7.850 \times 10^2$ | 206 | 0.3 | 600 |
| PE buoy | $3.000101 \times 10^1$ | 0.043 | 0.2 | n/a |
| PV panel | $2.188 \times 10^2$ | 50 | 0.2 | n/a |

$$p_j^w = \frac{\sum_{i=1}^n p_i \times w_i^\alpha}{\sum_{i=1}^n w_i^\alpha} \tag{4}$$

$$w_i = 1 - \frac{d_i}{d_{max}} \tag{5}$$

The Korea Hydrographic and Oceanographic Administration [30] have provided the metocean data at the Saemangeum Seawall Lake. In this study, the environmental data for last three years was collected to determine the design regular wave with the maximum wave height of 1.5 m and a period of 18.9 s based on the 100 year-return period. A scale factor of 0.75 was applied to the panel pressure and acceleration RAOs because the wave amplitude ratio was 0.

In addition, the corresponding current and wind speeds were 0.32 m/s and 36.16 m/s as delineated in Table 7. The wind- and current-induced drag forces *F* and moments *M* were also taken into account using Equations (6) and (8), respectively. As shown in Equation (7), the drag coefficient $C_d$ of a flat surface with an angle of attack $\phi$ can be estimated based on a reference [31–33]. $v_x$, $v_y$, and $v_z$ in Equation (6) are velocity components of either current or wind, while $A_x$, $A_y$, and $A_z$ are projected area components of incident current or wind. As long as the distance vector *r* from the COM to a considered area is identified, drag moment can be drawn using Equation (8).

$$\boldsymbol{F} = \frac{1}{2} C_d \rho \left( A_x v_x^2 \boldsymbol{i} + A_y v_y^2 \boldsymbol{j} + A_z v_z^2 \boldsymbol{k} \right) \tag{6}$$

$$C_d = \begin{cases} 2\pi \tan \boldsymbol{\phi} \ for \ \boldsymbol{\phi} < 8^\circ \\ \frac{1}{0.222 + 0.283 \sin \boldsymbol{\phi}} \ for \ \boldsymbol{\phi} \ge 8^\circ \end{cases} \tag{7}$$

$$\boldsymbol{M} = \boldsymbol{r} \times \boldsymbol{F} \tag{8}$$

**Table 7.** Environmental conditions for structural frequency response analysis.

| Item | Value |
|---|---|
| Wave height (m) | 1.50 |
| Wave period (s) | 18.90 |
| Current speed (m/s) | 0.32 |
| Wind speed (m/s) | 36.16 |

Cases for the structural analyses are summarized in Table 8 according to the applied loads where $\rho_{sea}$, $g$, and $z$ are seawater density, gravity, and vertical location of wet structure. The wave and current directions were assumed to be the same, but the wind directions were 90° to maximize the wind-induced drag force and moment. Therefore, six structural analysis cases such as M1-S1, M1-S2, etc. were generated considering even and uneven mass distributions of M1 and M2.

**Table 8.** Cases for structural frequency response analysis.

| Case | Wave Dir (Deg) | Current Dir (Deg) | Wind Dir (Deg) | Gravity (G) | Buoyancy (N) |
|---|---|---|---|---|---|
| M1-S1 or M2-S1 | 0 | 0 | 90 | 1 | $\rho_{sea}gz$ |
| M1-S2 or M2-S2 | 45 | 45 | 90 | 1 | $\rho_{sea}gz$ |
| M1-S3 or M2-S3 | 90 | 90 | 90 | 1 | $\rho_{sea}gz$ |

*4.2. Structural Analysis Result*

Frequency response analyses were conducted using a commercial finite element analysis (FEA) code, Abaqus/Standard [34]. There were negligible unbalance forces and moments were witnessed with only hydrodynamic loads. This means that the load RAOs from the hydrodynamic frequency response analyses were normally transferred to the structural frequency response analysis models.

Unlike other commercial FEA codes, Abaqus allows one to view shear stresses in beam elements after assigning effective shear stiffness factor to each beam section. It is used to prevent the shear stiffness from becoming too large in slender beam elements. For the typical beam sections listed in the Abaqus beam library, the Timoshenko stiffness factors were used [34].

In this paper, normal stress magnitude $\sigma$ and shear stress magnitude $\tau$ were calculated using real components of $\sigma_r$ and $\tau_r$ and imaginary components of $\sigma_i$ and $\tau_i$ (refer to Equations (9) and (10)). Equation (11) was used to produce von Mises equivalent stress $\sigma_v$.

$$\sigma = \sqrt{\sigma_r^2 + \sigma_i^2} \tag{9}$$

$$\tau = \sqrt{\tau_r^2 + \tau_i} \tag{10}$$

$$\sigma_v = \sqrt{\sigma^2 + 3\tau^2} \tag{11}$$

Figures 8 and 9 include normal, shear, and von Mises stress distributions in the frame structures under uneven and even mass conditions. The uneven mass distribution involves not only extremely high stress concentration in way of the hinge joints as seen in Figure 8, but also less uniform displacement along the wave direction of 0°. Meanwhile, as depicted in Figure 9, the stresses were quite relieved and the relative rotations at the hinge joints were found to be following the incident wave pattern after the mass distribution was evenly adjusted.

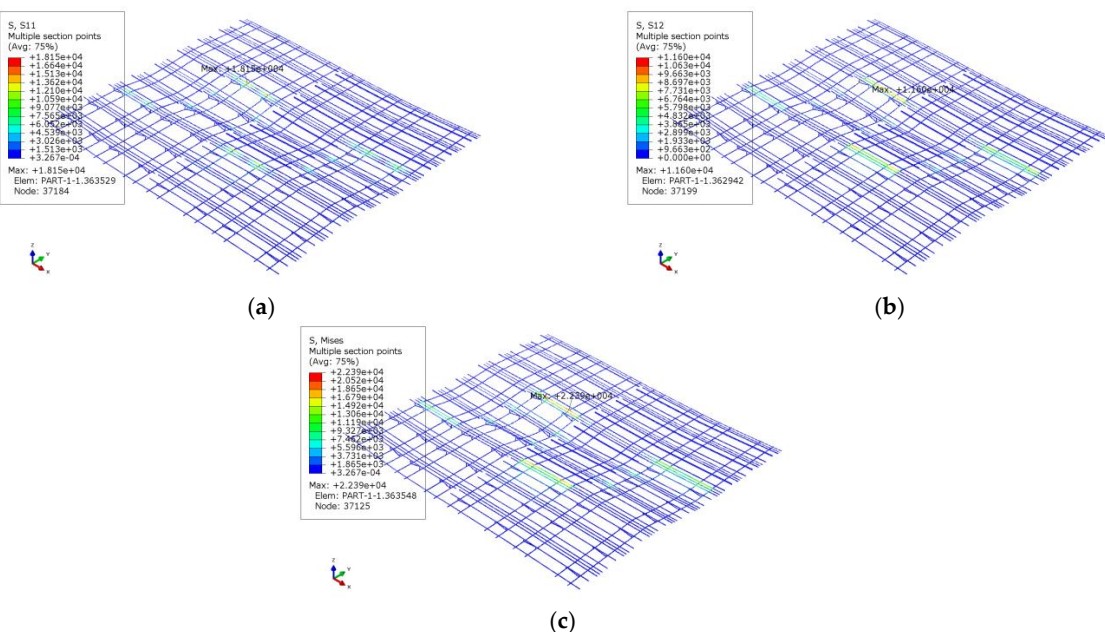

**Figure 8.** Stress distribution for the case M1-S1 (uneven mass and wave incident angle 0°): (**a**) normal stress; (**b**) shear stress; (**c**) von Mises stress.

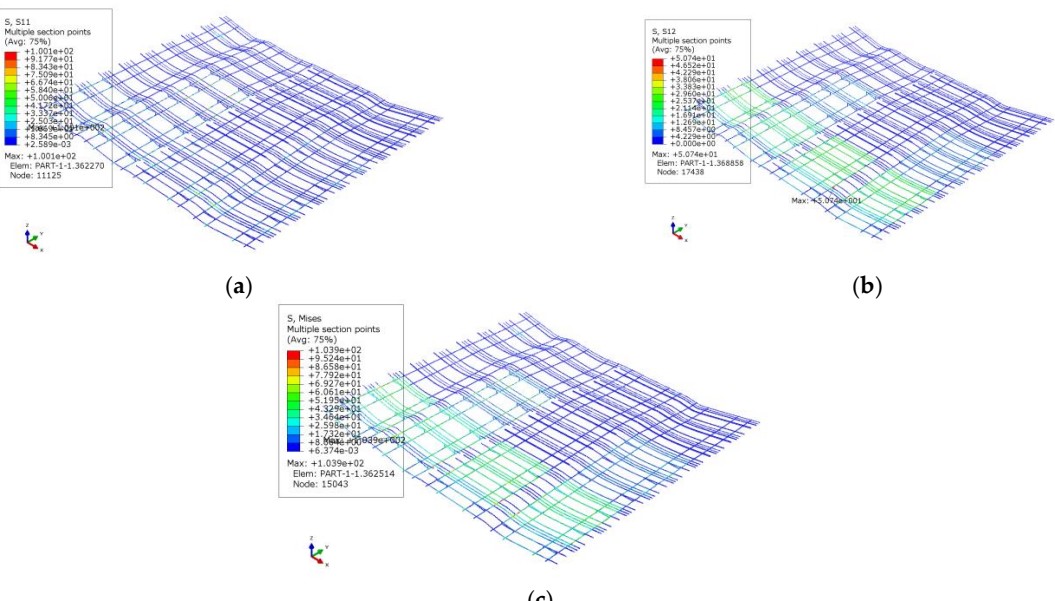

**Figure 9.** Stress distribution for the case M2-S1 (even mass and wave incident angle 0°): (**a**) normal stress; (**b**) shear stress; (**c**) von Mises stress.

The maximum von Mises stress in the FRP structure was reduced from 4436 MPa to 86 MPa after the even mass distribution (refer to Figure 10). The solid red cells in Figure 10 imply the floater unit where the maximum stress occurred. Very huge stresses in the POSMAC frame are found in Figures 8c and 11a. After mass distribution was changed uniformly between the floater units, these stresses were reduced to 339 MPa which is less than the yield strength of the POSMAC sheet of 600 MPa. The maximum von Mises stresses were summarized in Table 9. These dramatic reductions in the stresses were possible by equally modifying the distribution of mass between floaters.

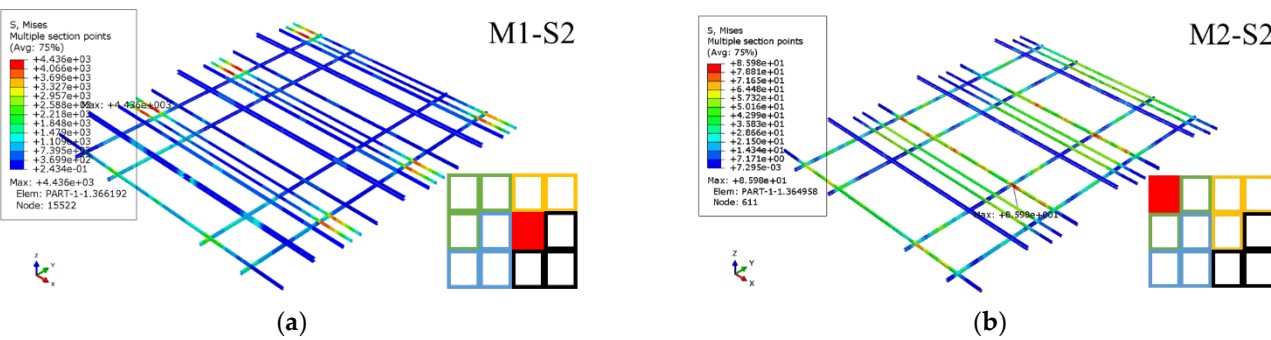

**Figure 10.** Maximum von Mises stresses in FRP frames: (**a**) M1-S2 (uneven mass and wave incident angle of 45°); (**b**) M2-S2 (even mass and wave incident angle of 45°).

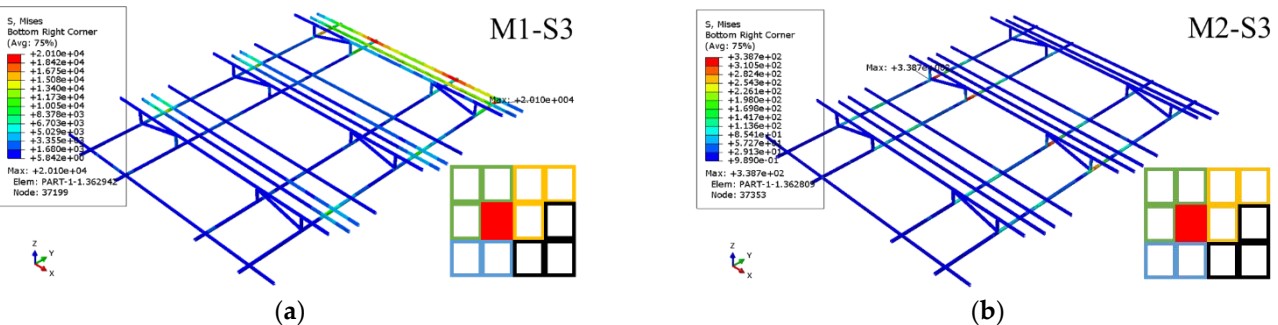

**Figure 11.** Maximum von Mises stresses in POSMAC frames: (**a**) M1-S3 (uneven mass and wave incident angle of 90°); (**b**) M2-S3 (even mass and wave incident angle of 90°).

**Table 9.** Maximum von Mises stress.

| Case | FRP (MPa) | POSMAC (MPa) |
|---|---|---|
| M1-S1 | 4754 | 22,390 |
| M1-S2 | 4436 | 20,550 |
| M1-S3 | 4485 | 20,100 |
| M2-S1 | 22 | 104 |
| M2-S2 | 86 | 288 |
| M2-S3 | 34 | 339 |

Even if the maximum von Mises stresses in the POSMAC frames became less than the yield strength, they were quite larger than those in the FRP frames. The POSMAC sheet is mechanically formed into the channel section, which is categorized into asymmetric open sections, thus laborers can assemble POSMAC beams into frame structures and dismantle them into beam members again. For this reason, asymmetric open sections have been preferred over closed sections or I-sections. From the point of view of structural safety, it can be concluded that the use of asymmetric open sections is not recommended.

## 5. Conclusions

In this study, the effect of the mass distribution between floating units was evaluated for a floating PV power plant with four sub-plants. The diffraction panel pressure and acceleration RAOs obtained from the hydrodynamic frequency response analyses were transferred to the finite element model through the distance-weighted method. The calculation procedure was called the spectral fatigue analysis, which was proposed in some classification societies for application to ships and offshore units [35–37]. The method had been applied to a range of marine structures, such as a jacket platform [38], wind turbines [39,40], floating offshore oil/gas unit [41], etc. The frequency response structural

analyses under the design wave, wind, and current based on 100-year metocean data yielded the following conclusions:

First, even though asymmetric open sections such as channels are preferred due to the convenience of in situ assembly of frame structures, they would lead to very high bending and shear stresses and be less commonly recommended than symmetric open sections of I-shape. Second, mass distributions in multiple floating units must be controlled equally and evenly. Otherwise, there would be huge stresses by way of the hinge joints because of the shear loads due to relative displacements between successive floating units.

Not only is checking maximum stresses over yield strength important, but also, buckling stability should be verified through national or international standards [5,42,43]. It was difficult to carry out the buckling checks by those standards because the load RAOs which were in harmonic form should be presented in the quasi-static form. A new technique to cover the buckling code checks should be developed in the near future.

**Author Contributions:** Conceptualization, J.C.; methodology, J.C.; software, C.B.L.; validation, C.B.L. and J.C.; formal analysis, J.C. and C.B.L.; investigation, J.C.; resources, J.C.; data curation, J.C.; writing—original draft preparation, C.B.L.; writing—review and editing, C.B.L.; visualization, C.B.L.; supervision, J.C.; project administration, J.C.; funding acquisition, J.C. All authors have read and agreed to the published version of the manuscript.

**Funding:** This research was conducted with the supports from research grants from Hyundai Global and the Korea Energy Technology Evaluation and Planning funded by the Ministry of Trade, Industry and Energy of Korea (No. 20213000000030). Authors appreciated Weihai Research Institute of Wu-han University of Technology (WHYJY-KJ2021-009).

**Institutional Review Board Statement:** Not applicable.

**Informed Consent Statement:** Not applicable.

**Data Availability Statement:** Not applicable.

**Conflicts of Interest:** The authors declare no conflict of interest.

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
