# Peer review of "Structural Effects of Mass Distributions in a Floating Photovoltaic Power Plant"

_jmse, doi:10.3390/jmse10111738_

Round 1

Reviewer 1 Report

Relating to the response to my comments for v1:

1.  Also adding some data, the specifications are yet not sufficient to reproduce the results.

4. All the results must by justified, by applying acceptable assumptions validated by the common practice, or by verification as part of the study.  The response is not sufficient.

6.  As agreed by the authors, the design cannot be presented with no assessment of buckling, applying any design acceptable method.

Author Response

Reviewer #1

Please find them in the manuscript in purple words

Relating to the response to my comments for v1:

C1: Also adding some data, the specifications are yet not sufficient to reproduce the results.

A1: The more detailed specifications of sub-plants were described in Figure2(b)

C2: All the results must by justified, by applying acceptable assumptions validated by the common practice, or by verification as part of the study. The response is not sufficient.

A2: With funding from a private company, we are performing the experimental study for validating our numerical method in predicting the dynamic response of the multiconnected floating solar platform. A significant observation was also found in the experimental study. It will be published in our next more high-quality journal.

C3: As agreed by the authors, the design cannot be presented with no assessment of buckling, applying any design acceptable method.

A3: Not only checking of maximum stresses over yield strength is important, but also buckling stability should be verified through national or international standards [1-3]. It was difficult to carry out the buckling checks by those standards because the load RAOs which were in harmonic form should be presented in the quasi-static form. A new technique to cover the buckling code checks should be developed in near future. The sentences were added to the conclusion.

  1. American Institute of Steel Construction (AISC). Steel Construction Manual – Allowable Stress Design, 9th ed.; AISC: Chicago Illinois, USA,
  2. American Institute of Steel Construction (AISC), Specification for Structural Steel Buildings; AISC: Chicago Illinois, USA, 2015.
  3. European Committee for Standardization (CEN), Eurocode 3 Part 1.1-1.5; CEN: Brussels, Belgium, 2005.

Reviewer 2 Report

1.        Why put four different sub-plants together?

2.        Why the fiber rope mooring line can be used in 4 meters in this study? How many years can be hold for the line?

3.        The ABSTRACT needs to be modified. There are too many texts to describe the background and introduction on the floating photovoltaic power plant. And it is no use for “weight in each floating unit should be evenly controlled if hinged joints are used to link the floaters”. The key problem, main research contents, new method, new finding etc. should be presented in this part.

4.        The POSCO and POSMAC should be explained when it is firstly used. And consistent with the item in Line 106 Page 3.

5.        The authors should also find some rothers’ research, not only Korea researchers’.

6.        Line 61 Page 2, what does “ technological bac” mean?

7.        The INTRODUCTION part is badly prepared. There are some discontinuous descriptions on PV power plant, mooring. The most important point in this part is to demonstrate what is the problem, that will be solved in the following part.

8.        A little confused on “The configuration of a buoy is shown in Figure 2(d)”. The buoy is not found in the figure.

9.        Why can the potential flow theory be used here under “buoys and PV modules were considered to be a single rigid body”?

10.    Why choose so big range for the frequency (until 7rad/s) in the hydrodynamic analysis?

11.    Still confused on the hydrodynamic load transferred to FEA. Could the authors explain further?

12.    Why not make analysis with irregular waves?

13.    The references are suggested to be the recent 5 years journal paper, rather than reports and software manuals.

Author Response

Thanks

Reviewer 3 Report

The structural analysis of a moored FPV power plant was carried out under a range of wave frequencies. The configuration of the power plant comprises four sub-plants supported by three floaters. The sub-plants were hinged and commercial potential flow software was deployed to calculate the response of the power plant over a range of frequencies. There are four major flaws in the hydrodynamic analysis below which authors need to address them carefully.

1- Chosen range of frequencies and the water depth are not compliant. The range of frequencies partly falls on high-order wave theories as well as surpassing the wave break criteria and it is out of the range of validity of the hydrodynamic software.

2- There is no information on the mooring system characteristics.

3- There is no information to show the validity of the hydrodynamic model.

4- There is no information on the convolution of the frequency domain hydrodynamic response into an appropriate form for the structural analysis.

Round 2

Reviewer 1 Report

No comments.